chemical physics

inhibition, mild steel, EIS, theoretical calculation, acid corrosion

**Author for correspondence:**
Y. Chen
e-mail: chen123yu123@163.com

# Inhibition behaviour of mild steel by three new benzaldehyde thiosemicarbazone derivatives in 0.5 M H$_2$SO$_4$: experimental and computational study

H. H. Zhang[1,3], C. K. Qin[2], Y. Chen[1] and Z. Zhang[3]

[1]Department of Chemical Engineering and Safety, and [2]College of Aeronautical Engineering, Binzhou University, Binzhou, Shandong 256600, People's Republic of China
[3]Department of Chemistry, Zhejiang University, Hangzhou, Zhejiang 310027, People's Republic of China

YC, 0000-0002-4169-039X

Three new benzaldehyde thiosemicarbazone derivatives namely benzaldehyde thiosemicarbazone (BST), 4-carboxyl benzaldehyde thiosemicarbazone (PBST) and 2-carboxyl benzaldehyde thiosemicarbazone (OCT) were synthesized and their inhibition effects on mild steel corrosion in 0.5 M H$_2$SO$_4$ solution were studied systematically using gravimetric and electrochemical measurements. Weight loss results revealed that PBST exhibited the highest inhibition efficiency of 96.6% among the investigated compounds when the concentration was 300 µM. The analysis of polarization curves indicated that the three benzaldehyde thiosemicarbazone derivatives acted as mixed type inhibitors and PBST and OCT predominantly anodic. The adsorption process of all these benzaldehyde thiosemicarbazone derivatives on Q235 steel surface in 0.5 M H$_2$SO$_4$ solution conformed to Langmuir adsorption isotherm. Scanning electron microscopy was conducted to show the presence of benzaldehyde thiosemicarbazone derivatives on Q235 mild steel surface. The results of theoretical calculations were in good agreement with that of experimental measurements.

## 1. Introduction

Mild steel has been applied as a popular construction material in petroleum, food, chemical and engineering industries [1–3].

**Table 1.** Physical and chemical properties of the synthesized compounds.

| no. | molecular structure | abbreviations | structure characterizations |
|---|---|---|---|
| 1 | | BST | $C_8H_9N_3S$ (mol. wt. 179) |
| | | | M.P. 164–165°C |
| | | | IR spectrum (KBr, cm$^{-1}$) |
| | | | 3399, 3144, 1599, 1283, 1098 |
| 2 | | PBST | $C_9H_9O_2N_3S$ (mol. wt. 223) |
| | | | M.P. 197–199°C |
| | | | IR spectrum (KBr, cm$^{-1}$) |
| | | | 3482, 3148, 1613, 1277, 1092 |
| 3 | | OCT | $C_9H_9O_2N_3S$ (mol. wt. 223) |
| | | | M.P. 203–205°C |
| | | | IR spectrum (KBr, cm$^{-1}$) |
| | | | 3173, 1606, 1270, 1105 |

However, mild steel is easily corroded in acidic solutions when they are serving in industrial washing, acid de-scaling and oil well acidization [4–6], which may cause significant economic losses and security risks. The use of inhibitors to prevent or minimize the considerable damage of mild steel in acid environment has been found to be one of the most economical and efficient methods [7–10]. It is generally believed that organic compounds containing heteroatoms such as nitrogen, oxygen and sulfur, or their molecular structure containing heterocyclic rings or polar functional groups serve as excellent organic inhibitors in acidic media [10]. The reason is that these compounds can form a strong chemical bond with the metal on the solid/liquid surface through charge transfer [11–13] and block the active site on the mild steel surface, thereby resisting the corrosion of mild steel in corrosive environment [14–17].

Gravimetric measurements [18,19], potentiodynamic polarization curves [20–22] and electrochemical impedance spectroscopy (EIS) [23–26] are classical methods to evaluate the inhibition behaviour. Theoretical calculation is a powerful technique to establish the relationship between the inhibition behaviour and molecular structure [27–29], which is helpful for designing a more effective inhibitor molecule. Moreover, some useful parameters, including the energy of the highest occupied molecular orbital, the energy of the lowest unoccupied molecular orbital, the energy gap and dipole moment can supply important information about the inhibition mechanism.

The objective of the present work is to investigate the inhibition behaviour of Q235 mild steel in 0.5 M $H_2SO_4$ solution containing three new synthesized benzaldehyde thiosemicarbazone derivatives namely benzaldehyde thiosemicarbazone (BST), 4-carboxyl benzaldehyde thiosemicarbazone (PBST) and 2-carboxyl benzaldehyde thiosemicarbazone (OCT) (table 1). The reason of choosing these compounds is that, firstly, these compounds contain various adsorption centres including oxygen, nitrogen and sulfur heteroatoms and –NH$_2$, –OH functional groups. Secondly, benzaldehyde thiosemicarbazone exhibited high inhibition efficiency for iron-base metallic glassy alloy in 0.5 M $H_2SO_4$ solution at 30°C, as previous reported [30]. Thirdly, these inhibitors can be easily synthesized with high yield. The study was carried out using weight loss measurement, potentiodynamic polarization curves, EIS and scanning electron microscopy (SEM). The correlation between the inhibition efficiencies of different substitution on the thiosemicarbazone compounds is discussed. Moreover, theoretical calculations were conducted to evaluate the inhibition mechanism.

# 2. Methods

## 2.1. Materials

The studied benzaldehyde thiosemicarbazone derivatives were synthesized according to the literature [30,31] and the chemical reaction equation is shown in figure 1. Table 1 depicts the physical and chemical properties of the synthesized compounds.

**Figure 1.** Chemical reaction equation of the studied compounds.

**Table 2.** Chemical composition (mass fraction, wt%) of Q235 mild steel samples.

| C | Mn | Si | S | P | Fe |
|---|---|---|---|---|---|
| 0.16 | 0.53 | 0.30 | <0.055 | <0.045 | Bal. |

## 2.2. Weight loss measurements

Weight loss experiments were performed in 500 ml 0.5 M $H_2SO_4$ solution containing different concentrations of benzaldehyde thiosemicarbazone derivatives. The testing time is 8 h at 298 K. Square specimens of Q235 carbon steel having dimensions $50 \times 25 \times 5$ mm were used for the gravimetric tests. The chemical composition of Q235 carbon steel is shown in table 2. The Q235 specimens were accurately weighed after degreasing with acetone and drying in $N_2$. After 8 h corroding time, the Q235 samples were moved out and the surface was scrubbed with a bristle brush, and then weighed again. For each case, at least triplicate experiments were conducted and the average results are reported.

The corrosion rate for Q235 mild steel was derived from the following expression [32]:

$$CR = \frac{87.6 \times W}{A \times t \times \rho},$$

(2.1)

where $W$ is the mass loss of Q235 mild steel without and with addition of inhibitors in milligrams, $A$ equals 32.5 cm$^2$ in our experimental condition, $t$ is the testing time of 8 h, $\rho$ is the Q235 mild steel density of $7.86 \times 10^3$.

Thus, the inhibition efficiency (IE%) and surface coverage ($\theta$) can be obtained from the corrosion rate using the following equation [32]:

$$IE(\%) = \left(1 - \frac{CR_{inhi}}{CR_{free}}\right) \times 100$$

(2.2)

and

$$\theta = 1 - \frac{CR_{inhi}}{CR_{free}},$$

(2.3)

where $CR_{inhi}$ and $CR_{free}$ are the obtained corrosion rates of Q235 mild steel with and without benzaldehyde thiosemicarbazone derivatives in 0.5 M $H_2SO_4$ solution, respectively.

## 2.3. Potentiodynamic polarization studies

The electrochemical measurements were carried out in a cylindrical glass cell of 250 ml with traditional three electrodes. A saturated calomel electrode (SCE) and a large platinum foil were employed as reference and counter electrode, respectively. Cylindrical Q235 mild steel sealed with Teflon was used

as working electrode and the exposed area was 0.50 cm$^2$. The exposed surface was abraded with fine sand paper and then polished to mirror using 2.5 μm diamond paste, cleaned with double-distilled water and finally immersed into the electrochemical glass cell containing 0.5 M H$_2$SO$_4$ solution without and with different concentrations of benzaldehyde thiosemicarbazone derivatives for at least 1 h. When the open circuit potential ($E_{ocp}$) reached a steady state, the potentiodynamic polarization study was performed using CHI660A electrochemical workstation at a scan rate of 1 mV s$^{-1}$. The scanning potential ranges from $E_{ocp} - 250$ mV to $E_{ocp} + 250$ mV. Tafel extrapolation method was employed to obtain some useful parameters, including corrosion potential ($E_{corr}$), corrosion current density ($j_{corr}$), anodic and cathodic Tafel slopes. The inhibition efficiency $\eta_P$ (%) is then derived from the corrosion current density as follows:

$$\eta_P\% = \left(1 - \frac{j_{inhi}}{j_{free}}\right) \times 100\%, \tag{2.4}$$

where $j_{inhi}$ and $j_{free}$ are the obtained corrosion current densities with and without BST, PBST and OCT inhibitors, respectively.

## 2.4. Electrochemical impedance experiments

EIS was conducted using PARSTAT 2273 measurement unit in the frequency range of 10 mHz–100 kHz at $E_{ocp}$ and the scanning always initiated from high frequency to low frequency. The voltage amplitude was 5 mV. Each concentration was repeatedly tested three times or more and the average results were calculated. The EIS experimental data was analysed using Z-View software. The inhibition efficiency $\eta_{EIS}$ (%) can be obtained from the charge transfer resistance as follows:

$$\eta_{EIS}\% = \frac{R_{ct} - R_{ct}^0}{R_{ct}}, \tag{2.5}$$

where $R_{ct}^0$ and $R_{ct}$ are the charge transfer resistance for Q235 mild steel in uninhibited and inhibited solution, respectively.

## 2.5. Scanning electron microscopy measurements

The Q235 mild steel surface after corroded in 0.5 M H$_2$SO$_4$ solution at 298 K without and with inhibitors was observed with SEM model Hitachi SU80 instrument at an accelerating voltage of 5 kV at 2000× magnification. EDX detector model coupled with SEM was used to evaluate the surface composition of Q235 mild steel.

## 2.6. Theoretical calculations

As described in previous literature [33,34], the geometric optimizations of the synthesized derivatives and quantum chemical calculations were performed using the functional hydride B3LYP density functional theory (DFT) formalism. During the calculations, the electron basis set 6-31G (d, p) in the standard Gaussian-03 software package was employed. As a result, some useful quantum chemical parameters, such as energy of the lowest unoccupied molecular orbital ($E_{LUMO}$), the energy of the highest occupied molecular orbital ($E_{HOMO}$), the energy gap ($\Delta E$) between LUMO and HOMO, the ionization potential ($I$), the electron affinity ($A$), dipole moment, the global hardness ($\eta$) and the global softness ($\sigma$) were calculated.

# 3. Results and discussion

## 3.1. Weight loss tests

The inhibitive effect of the synthesized benzaldehyde thiosemicarbazone derivatives (BST, PBST and OCT) for Q235 mild steel in 0.5 M H$_2$SO$_4$ solution at 298 K was initially investigated with weight loss measurements. The calculated values of corrosion rate, inhibition efficiency and surface coverage are summarized in table 3. Obviously, the corrosion rate decreased considerably with addition of these compounds compared to the blank, which may be related to the strong adsorption of these compounds onto Q235 mild steel surface and forming a protective physical barrier to resist the acid

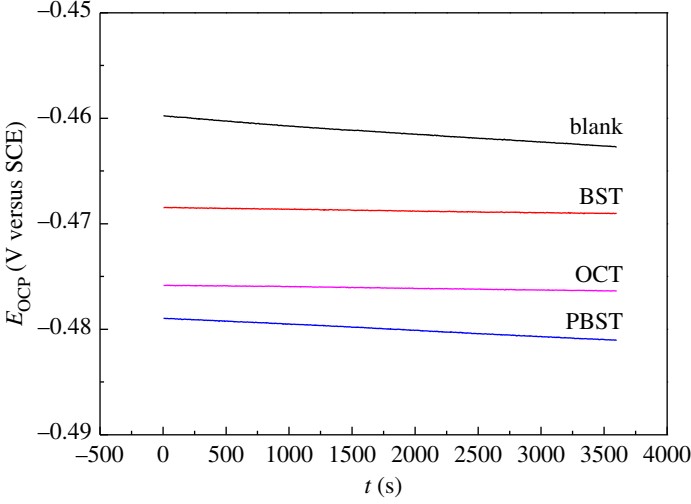

**Figure 2.** Open circuit potential for Q235 mild steel in 0.5 M H$_2$SO$_4$ solution without and with 300 µM BST, PBST and OCT inhibitor.

**Table 3.** The weight loss parameters for Q235 steel in 0.5 M H$_2$SO$_4$ containing different concentrations of BST, PBST and OCT inhibitor at 298 K.

| inhibitor | $C_{inh}$ (µM) | CR (mg cm$^{-2}$ h$^{-1}$) | $\eta$ (%) | $\theta$ |
|---|---|---|---|---|
| blank | 0 | 6.09 | — | |
| BST | 50 | 2.75 | 54.8 | 0.548 |
| | 100 | 1.71 | 71.9 | 0.719 |
| | 200 | 1.12 | 81.6 | 0.816 |
| | 300 | 0.87 | 85.7 | 0.857 |
| PBST | 50 | 2.33 | 61.7 | 0.617 |
| | 100 | 1.24 | 79.6 | 0.796 |
| | 200 | 0.66 | 89.2 | 0.892 |
| | 300 | 0.21 | 96.6 | 0.966 |
| OCT | 50 | 2.11 | 65.4 | 0.654 |
| | 100 | 1.37 | 77.5 | 0.775 |
| | 200 | 0.76 | 87.5 | 0.875 |
| | 300 | 0.40 | 93.4 | 0.934 |

attack [35]. It is clear that with increasing the inhibitor concentration, the inhibition efficiency also increased and the highest value was found to be 85.7%, 96.6% and 93.4% for BST, PBST and OCT compounds, respectively at 300 µM, suggesting that these inhibitors effectively inhibited the Q235 mild steel corrosion in acidic medium. It can also be deduced from table 3 that at the same concentration, the inhibition efficiency follow the order: PBST > OCT > BST, indicating that PBST exhibits the best inhibitive performance compared to other two benzaldehyde thiosemicarbazone derivatives. This result may be correlated with its molecular structure of –COOH functional group at the $\rho$-substitution (table 1).

## 3.2. Open circuit potential curves

The open circuit potential for Q235 mild steel in 0.5 M H$_2$SO$_4$ solution without and with 300 µM BST, PBST and OCT inhibitor is depicted in figure 2. It can be seen that the OCP reached a steady state after 1 h immersion time. Apparently, the OCP value moved in the negative direction with addition of 300 µM BST, PBST and OCT inhibitor compared to the blank. This shift may be correlated to the adsorption of these compounds on mild steel surface.

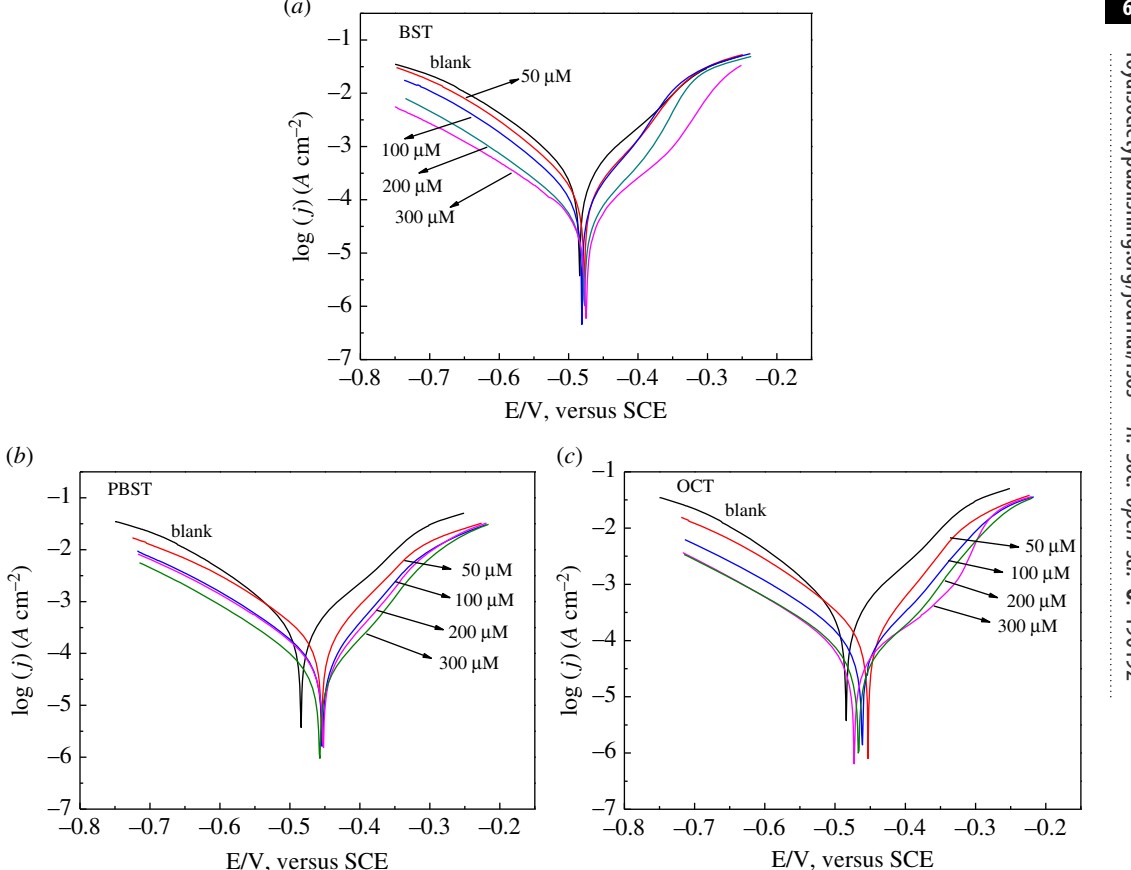

**Figure 3.** Polarization curves for Q235 mild steel in 0.5 M $H_2SO_4$ solution containing different concentrations of (*a*) BST, (*b*) PBST and (*c*) OCT inhibitor.

## 3.3. Potentiodynamic polarization curves

Figure 3 shows the polarization curves for Q235 mild steel in 0.5 M $H_2SO_4$ solution containing different concentrations of BST, PBST and OCT inhibitor. Apparently, the current densities of both the anodic and cathodic branches decreased with addition of benzaldehyde thiosemicarbazone derivatives, indicating that both the anodic and cathodic reaction rates were resisted which was generally due to the adsorption of these inhibitors at the active sites on the surface. It is noticeable that the shape of polarization curves without inhibitors is similar to that with addition of these three inhibitors. This phenomenon demonstrated that the addition of these inhibitors did not change the corrosion mechanism of Q235 mild steel dissolution in 0.5 M $H_2SO_4$ solution [36] and the inhibitive effect of these inhibitors is originated from the coverage of inhibitor molecules at the active sites to restrain their exposure to the acidic environment. In addition, it can be seen that there was no obvious trend observed in the $E_{corr}$ values for BST inhibitor compared to the blank, which moves to the positive direction with less than 85 mV in the presence of PBST and OCT inhibitors, suggesting that these inhibitors were of mixed type and PBST and OCT predominantly anodic [37–39]. The values of $E_{corr}$, $j_{corr}$, cathodic and anodic Tafel slopes, and $\eta_P$ (%) are listed in table 4. Obviously, $j_{corr}$ values decreased remarkably with addition of these inhibitors compared to the uninhibited. With increasing inhibitors concentration from 0 to 300 µM, the values of $j_{corr}$ decreased from 375.8 µA cm$^{-2}$ to 50.2, 34.8 and 36.1 µA cm$^{-2}$ for BST, PBST and OCT inhibitors, respectively. Therefore, $\eta_P$ exhibited a maximum value of 87.1%, 90.7% and 90.4% for BST, PBST and OCT inhibitor, respectively. Observation of table 4 shows that the inhibition efficiency obeys the order: PBST > OCT > BST, which is in good accordance with the gravimetric tests.

## 3.4. Electrochemical impedance spectroscopy measurements

The representative Nyquist plots for Q235 mild steel dissolution in 0.5 M $H_2SO_4$ solution at 298 K in the absence and presence of different concentrations of BST, PBST and OCT inhibitors are shown in figure 4. It is apparent that the Nyquist plots were considerably influenced after the addition of these

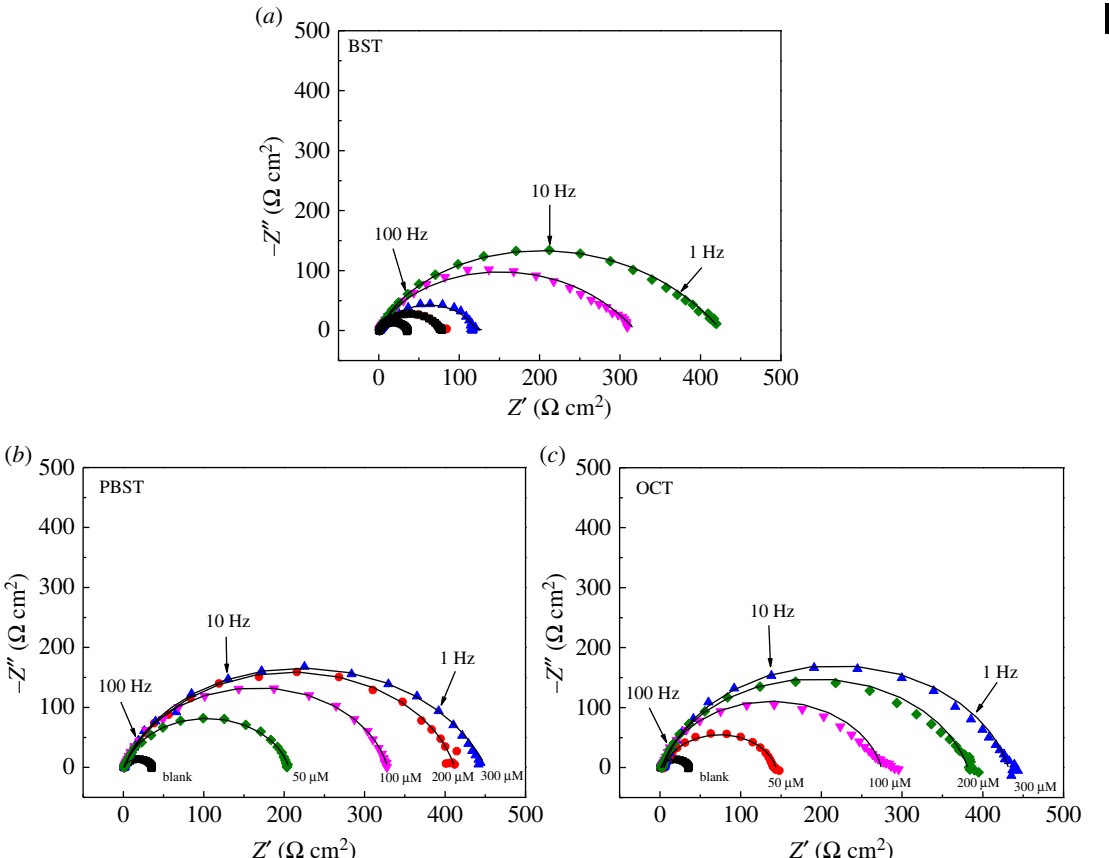

**Figure 4.** Nyquist plots for Q235 mild steel corrosion in 0.5 M H$_2$SO$_4$ solution at 298 K containing different concentrations of (*a*) BST, (*b*) PBST and (*c*) OCT.

**Table 4.** Polarization parameters for Q235 mild steel corroded in 0.5 M H$_2$SO$_4$ containing different concentrations of BST, PBST and OCT inhibitor at 298 K.

| inhibitor | $C_{inh}$ (µM) | $E_{corr}$ (mV) | $\beta_a$ (mV dec$^{-1}$) | $-\beta_c$ (mV dec$^{-1}$) | $j_{corr}$ (µA cm$^{-2}$) | $\eta_P$ (%) |
|---|---|---|---|---|---|---|
| blank | 0 | −479.1 | 113.9 | 102.4 | 375.8 | |
| BST | 50 | −477.2 | 119.5 | 105.0 | 218.5 | 41.9 |
| | 100 | −479.9 | 121.5 | 102.6 | 122.9 | 67.3 |
| | 200 | −478.8 | 116.7 | 97.7 | 54.1 | 85.6 |
| | 300 | −475.6 | 109.0 | 100.2 | 50.2 | 87.1 |
| PBST | 50 | −457.1 | 102.9 | 110.1 | 167.3 | 55.5 |
| | 100 | −448.6 | 105.2 | 108.3 | 67.2 | 82.1 |
| | 200 | −458.8 | 98.6 | 110.4 | 51.9 | 86.2 |
| | 300 | −460.4 | 100.4 | 108.2 | 34.8 | 90.7 |
| OCT | 50 | −456.2 | 98.4 | 126.8 | 198.6 | 47.1 |
| | 100 | −450.6 | 102.1 | 124.3 | 90.6 | 75.9 |
| | 200 | −455.8 | 98.3 | 112.9 | 53.0 | 85.8 |
| | 300 | −459.3 | 100.9 | 121.5 | 36.1 | 90.4 |

inhibitors into 0.5 M H$_2$SO$_4$ solution, which diameter was greater than that in the blank solution, suggesting that the Q235 mild steel dissolution process was remarkably restrained by these inhibitors. As observed, for the uninhibited case, the impedance spectrum contains only one depressed capacitive loop, while after the addition of BST, PBST and OCT inhibitors, the Nyquist plots show

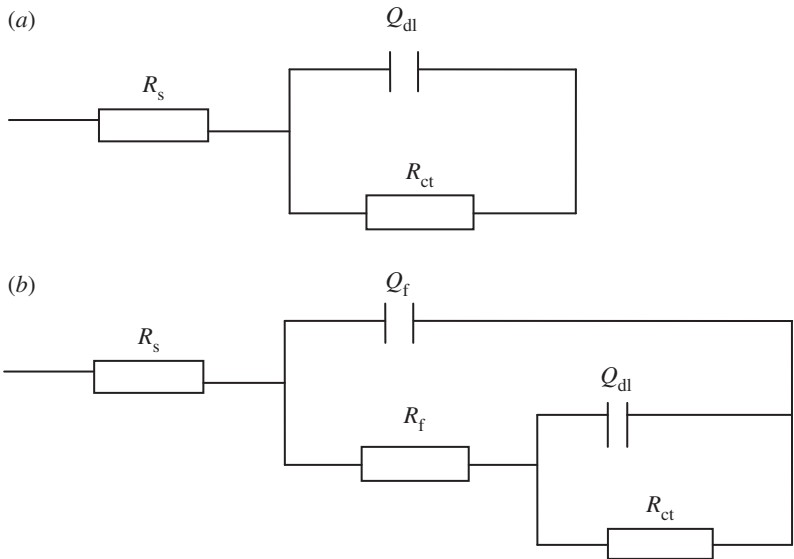

**Figure 5.** EEC model used to simulate the EIS data. ($R_s$: solution resistance, $Q_{dl}$: double layer capacitance; $R_{ct}$: charge transfer resistance; $Q_f$: film capacitance; $R_f$: film resistance).

**Table 5.** EIS parameters for Q235 mild steel in 0.5 M $H_2SO_4$ containing different concentrations of BST, PBST and OCT inhibitor.

| | $C_{inh}$ (μM) | $R_s$ ($\Omega$ cm$^2$) | $Q_f$ ($\Omega^{-1}$ s$^n$ cm$^{-2}$) | $R_f$ ($\Omega$ cm$^2$) | $R_{ct}$ ($\Omega$ cm$^2$) | $Q_{dl}$ ($\Omega^{-1}$ s$^n$ cm$^{-2}$) | $n_{dl}$ | $\eta_{EIS}$ (%) |
|---|---|---|---|---|---|---|---|---|
| blank | 0 | $0.88 \pm 0.02$ | | | $33.4 \pm 1.1$ | $175 \pm 6.1$ | 0.968 | — |
| BST | 50 | $1.05 \pm 0.03$ | $242 \pm 10$ | $2.16 \pm 0.08$ | $78.4 \pm 1.8$ | $84.6 \pm 4.3$ | 1 | 57.4 |
| | 100 | $0.99 \pm 0.03$ | $206 \pm 8$ | $2.44 \pm 0.04$ | $126.9 \pm 3.2$ | $72.2 \pm 2.0$ | 1 | 73.7 |
| | 200 | $1.02 \pm 0.02$ | $169 \pm 4$ | $12.1 \pm 0.16$ | $306.4 \pm 8.3$ | $56.6 \pm 2.2$ | 0.945 | 89.1 |
| | 300 | $1.02 \pm 0.03$ | $132 \pm 5$ | $16.8 \pm 0.12$ | $408.9 \pm 7.6$ | $51.4 \pm 3.1$ | 0.922 | 91.8 |
| PBST | 50 | $1.08 \pm 0.02$ | $213 \pm 7$ | $9.24 \pm 0.14$ | $175.6 \pm 5.2$ | $73.6 \pm 3.5$ | 0.948 | 80.9 |
| | 100 | $0.93 \pm 0.02$ | $182 \pm 5$ | $11.2 \pm 0.16$ | $252.3 \pm 4.9$ | $71.4 \pm 2.4$ | 0.916 | 86.7 |
| | 200 | $0.96 \pm 0.03$ | $130 \pm 3$ | $13.6 \pm 0.12$ | $371.6 \pm 8.2$ | $46.4 \pm 1.3$ | 0.953 | 91.0 |
| | 300 | $0.89 \pm 0.02$ | $114 \pm 2$ | $17.4 \pm 0.23$ | $485.3 \pm 7.1$ | $32.2 \pm 0.9$ | 0.942 | 93.2 |
| OCT | 50 | $0.84 \pm 0.03$ | $225 \pm 6$ | $3.58 \pm 0.11$ | $113.4 \pm 2.6$ | $84.6 \pm 1.7$ | 0.974 | 70.5 |
| | 100 | $1.03 \pm 0.02$ | $196 \pm 4$ | $6.26 \pm 0.15$ | $242.2 \pm 5.6$ | $58.2 \pm 1.3$ | 0.906 | 86.2 |
| | 200 | $1.06 \pm 0.02$ | $143 \pm 3$ | $11.4 \pm 0.12$ | $356.5 \pm 8.2$ | $41.2 \pm 1.2$ | 0.930 | 90.6 |
| | 300 | $0.97 \pm 0.03$ | $128 \pm 4$ | $15.7 \pm 0.14$ | $460.8 \pm 11$ | $24.8 \pm 0.8$ | 0.961 | 92.8 |

two capacitive loops which may correspond to the double electric layer and film capacitance, respectively. Obviously, the diameter of the capacitive loops became larger with the increase of inhibitor concentration. Figure 3 also shows that the centres of the impedance loops are below the real axis. This finding indicates a non-ideal electrochemical behaviour at the metal/solution interface [40,41], which generally resulted from the surface roughness and heterogeneities [42,43]. Therefore, a constant phase element CPE ($Q$) was used to replace capacity [44] and the thus improved electrochemical equivalent circuit (EEC) is depicted in figure 4, which was used to analyse the EIS data. The admittance of a CPE can be calculated using the following expression [45]:

$$Y_{CPE} = Y_0(j\omega)^n, \tag{3.1}$$

where $Y_0$ is the magnitude, $j$ equals $-1$, $\omega$ is the angular frequency, and $n$ is the phase shift, representing the surface inhomogeneity [46] (figure 5).

The fitted results are summarized in table 5. It can be seen that the values of $R_{ct}$ increased upon rising of inhibitors concentration, suggesting a higher corrosion resistance by the adsorption of these inhibitors onto

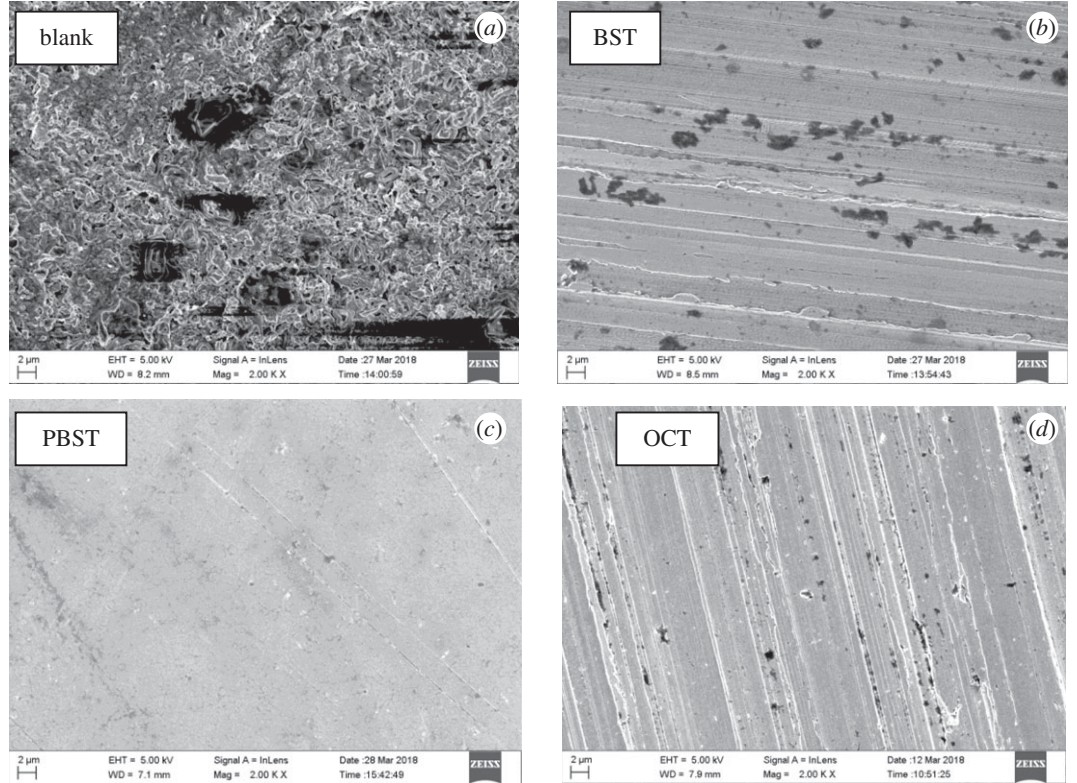

**Figure 6.** Surface morphology of Q235 mild steel after being corroded in 0.5 M $H_2SO_4$ solution at 298 K in the absence (*a*) and presence of 300 µM (*b*) BST, (*c*) PBST and (*d*) OCT.

Q235 mild steel surface. The $R_{ct}$ values reached 408.9, 485.3 and 460.8 $\Omega$ cm$^2$ for BST, PBST and OCT inhibitors respectively, when their concentration was 300 µM. Accordingly, the inhibition efficiency exhibited a maximum value of 91.8%, 93.2% and 92.8% for BST, PBST and OCT, respectively. Oppositely, the $Q_{dl}$ values decreased with the increase of inhibitors concentration. The reason may be that the synthesized benzaldehyde thiosemicarbazone derivatives adsorbed on the metal surface and formed a stronger chemical bond with steel than water molecules, thus the previously absorbed water molecules were replaced [47,48]. Additionally, the values of $n$ were all near to 1 in the absence and presence of these inhibitors (table 5), suggesting the homogeneous nature of the surface. Moreover, it is worth noting that the inhibition efficiencies of PBST and OCT inhibitor are higher than that of BST at the same concentration, which may be correlated to the presence of –COOH functional group in the molecular structure.

## 3.5. Surface investigation

SEM images of Q235 mild steel samples after being corroded in 0.5 M $H_2SO_4$ solution without and with addition of 300 µM BST, PBST and OCT inhibitors are shown in figure 6. It is observed that the Q235 mild steel surface was strongly damaged without inhibitors (figure 6*a*), while the surface was smooth and compact when 300 µM inhibitors was added (figure 6*b–d*), indicating that benzaldehyde thiosemicarbazone derivatives formed a protective physical barrier on the mild steel surface and retarded the aggressive acid attack. The presence of these inhibitors was further confirmed by EDX spectra, as shown in figure 7. It is worth noting that no characteristic peaks for nitrogen (N) and sulfur (S) can be found in the uninhibited solution (figure 7*a*), whereas both of them appeared on Q235 mild steel surface in 0.5 M $H_2SO_4$ solution containing 300 µM inhibitors (figure 7*b–d*), which indicated the presence of these inhibitor molecules to form a protective film on the Q235 mild steel surface. Moreover, the percentage atomic contents of elements obtained from EDX measurements for Q235 mild steel samples in the absence and presence of 300 µM BST, PBST and OCT is given in table 6. It also can be seen that the percentage atomic content of Fe reduced sharply with addition of 300 µM BST, PBST and OCT inhibitors compared to the blank, which was due to the surface coverage of these inhibitor molecules on Q235 mild steel surface.

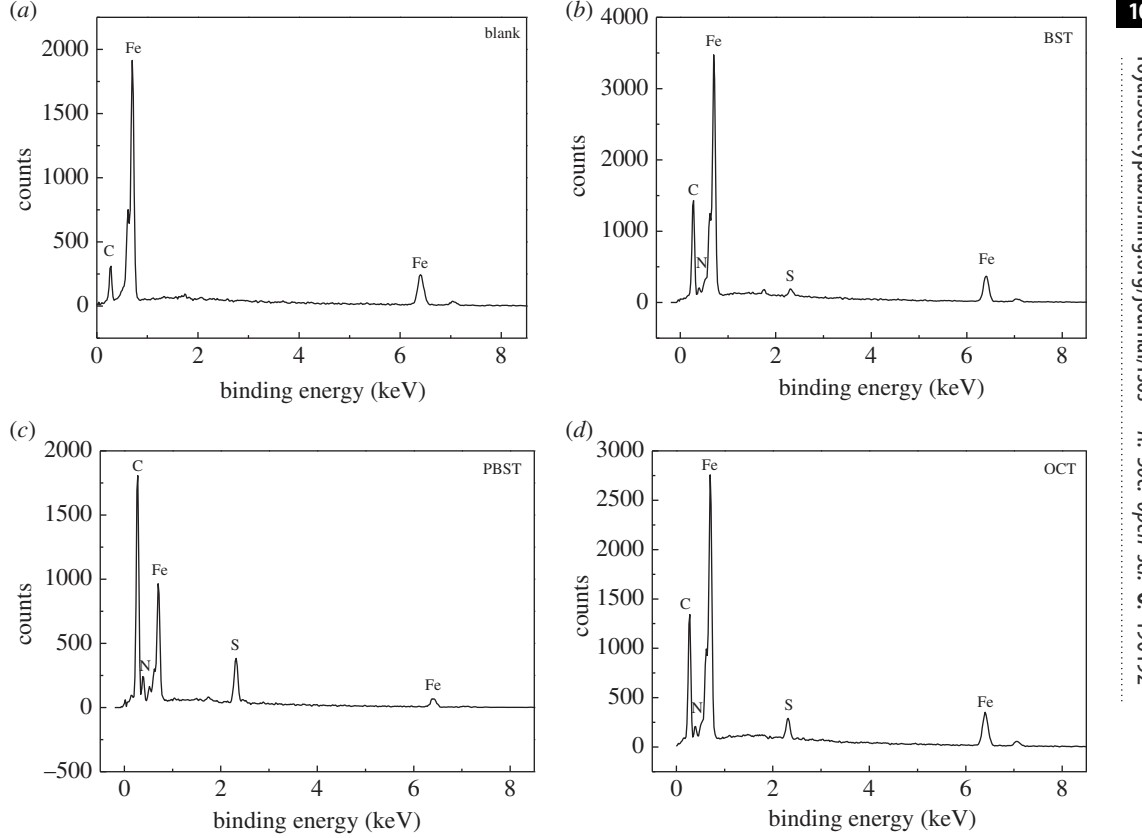

**Figure 7.** EDX spectra of Q235 mild steel surface in 0.5 M $H_2SO_4$ at 298 K in the absence (*a*) and presence of 300 µM (*b*) BST, (*c*) PBST and (*d*) OCT.

**Table 6.** EDX spectra results for mild steel samples in the absence and presence of 300 µM BST, PBST and OCT.

| inhibitor | Fe | C | N | S |
|---|---|---|---|---|
| blank | 70.78 | 29.22 | — | — |
| BST | 67.84 | 27.53 | 2.82 | 1.81 |
| PBST | 62.54 | 26.52 | 6.64 | 4.30 |
| OCT | 64.65 | 26.65 | 5.07 | 3.63 |

## 3.6. Adsorption isotherm

To further explore the adsorption mechanism of BST, PBST and OCT inhibitors onto Q235 mild steel in 0.5 M $H_2SO_4$ solution, different adsorption isotherms, including Langmuir, Flory-Huggins, Temkin, Freundlich and Frumkin isotherm models were employed. In the present study, a linear relationship between $c/\theta$ values and inhibitors concentration $c$ was established, as shown in figure 8, which indicated that the adsorption of these inhibitors on Q235 mild steel surface in 0.5 M $H_2SO_4$ solution conformed to Langmuir adsorption isotherm with the following expression [49,50]:

$$\frac{c}{\theta} = \frac{1}{K_{ads}} + c, \tag{3.2}$$

where $K_{ads}$ is the equilibrium constant of inhibitors adsorption onto Q235 mild steel surface, which values can be obtained from the intercept of figure 8. According to the relationship between the standard free energy of adsorption $\Delta G^0_{ads}$ and $K_{ads}$, $\Delta G^0_{ads}$ can be calculated from the value of $K_{ads}$ using the following equation [51]:

$$K_{ads} = \frac{1}{55.5} \exp\left(\frac{-\Delta G^0_{ads}}{RT}\right), \tag{3.3}$$

where $R$ is the molar gas constant and $T$ is the absolute temperature. The calculated equilibrium constant $K_{ads}$ and standard free energy of adsorption $\Delta G^0_{ads}$ are summarized in table 7.

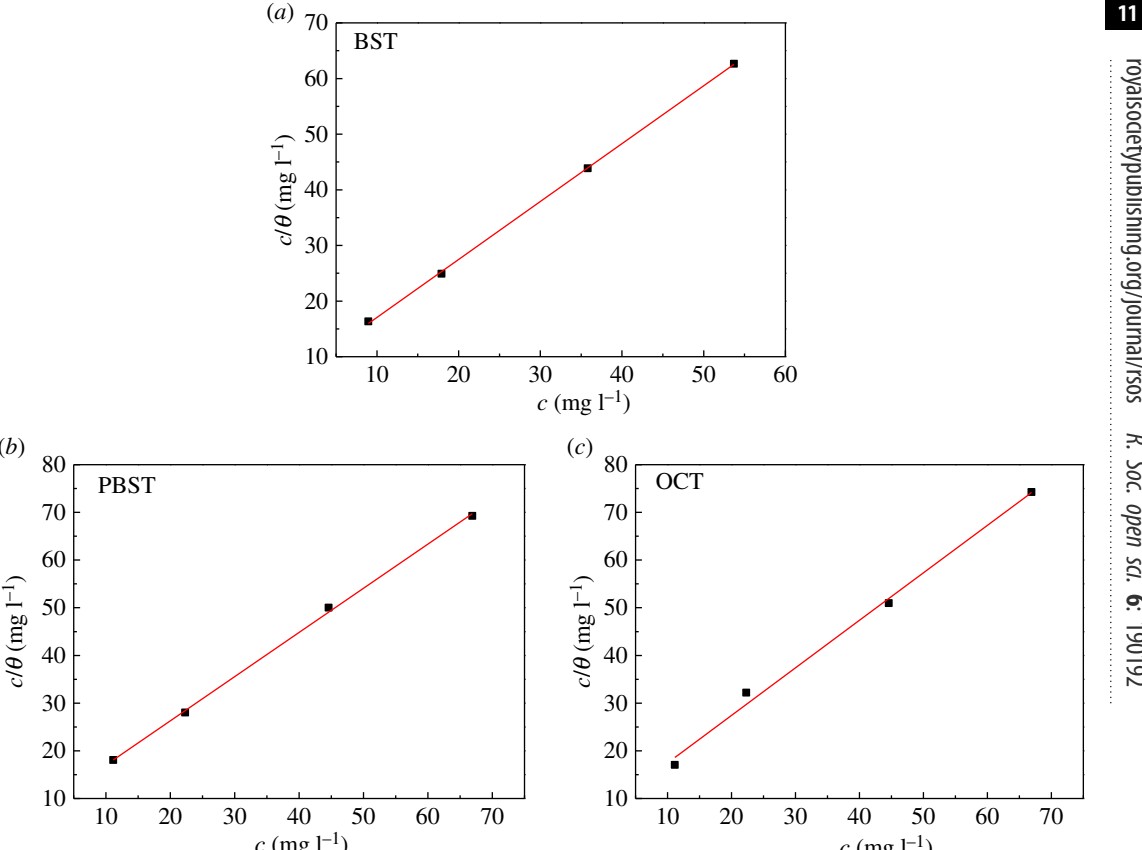

**Figure 8.** Langmuir isotherm for adsorption of (a) BST, (b) PBST and (c) OCT molecules onto Q235 mild steel in 0.5 M $H_2SO_4$ solution.

**Table 7.** The values of $K_{ads}$ and $\Delta G^0_{ads}$ for Q235 mild steel in the presence of BST, PBST and OCT inhibitors in 0.5 M $H_2SO_4$ solution.

| inhibitor | $K_{ads}(\times 10^4/M)$ | $\Delta G^0_{ads}$ (kJ mol$^{-1}$) | slope | $R^2$ |
|---|---|---|---|---|
| BST | 2.62 | −35.1 | 1.04 | 0.999 |
| PBST | 3.26 | −35.7 | 0.98 | 0.999 |
| OCT | 2.97 | −35.4 | 1.00 | 0.992 |

In our present measurements, the values of $\Delta G^0_{ads}$ are found to be −35.1, −35.7 and −35.4 kJ mol$^{-1}$ for BST, PBST and OCT inhibitors, respectively. It is reported that the adsorption of organic inhibitor molecules onto metal surface follows physical adsorption through electrostatic interaction when the value of $\Delta G^0_{ads}$ was positive to −20 kJ mol$^{-1}$, which conforms to chemisorptions involving charge sharing or charge transfer between the metal surface and inhibitor molecules when $\Delta G^0_{ads}$ value was negative to −40 kJ mol$^{-1}$ [52–54]. Therefore, it is reasonable to deduce that the adsorption process of BST, PBST and OCT inhibitors onto Q235 mild steel surface is a combination of both chemisorptions and physisorption, which are predominant chemisorptions.

## 3.7. Effect of temperature

To further obtain the thermodynamic and activation parameters, weight loss experiments were performed at different temperatures ranging from 25°C to 55°C. The calculated corrosion rate and inhibition efficiency under different temperatures are listed in table 8. It is apparent that the corrosion rate increases with raising temperature for all these inhibitors. Meanwhile, the inhibition efficiency increases with temperature as well, which corresponds to the chemisorptions mechanism of inhibitor molecule onto metal surface. This phenomenon has been explained by the specific interaction between

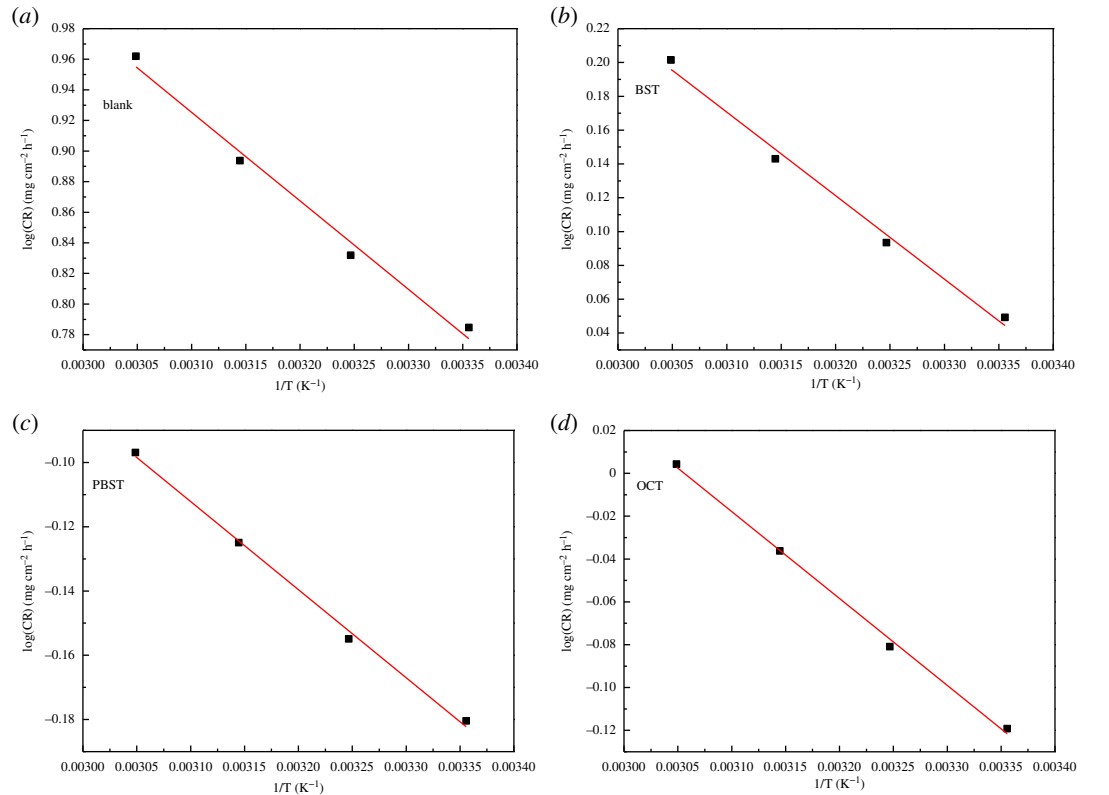

**Figure 9.** Log CR versus 1/T for Q235 mild steel in 0.5 M H$_2$SO$_4$ solution.

**Table 8.** Weight loss parameters for Q235 mild steel corroded in 0.5 M H$_2$SO$_4$ solution with addition of 200 μM BST, PBST and OCT inhibitors at different temperatures.

| inhibitor | temperature (°C) | blank CR (mg cm$^{-2}$ h$^{-1}$) | 200 μM CR (mg cm$^{-2}$ h$^{-1}$) | $\eta$ (%) |
|---|---|---|---|---|
| BST | 25 | 6.09 | 1.12 | 81.6 |
| | 35 | 6.79 | 1.24 | 81.7 |
| | 45 | 7.83 | 1.39 | 82.2 |
| | 55 | 9.16 | 1.59 | 82.6 |
| PBST | 25 | 6.09 | 0.66 | 89.2 |
| | 35 | 6.79 | 0.70 | 89.7 |
| | 45 | 7.83 | 0.75 | 90.4 |
| | 55 | 9.16 | 0.80 | 91.3 |
| OCT | 25 | 6.09 | 0.76 | 87.5 |
| | 35 | 6.79 | 0.83 | 87.8 |
| | 45 | 7.83 | 0.92 | 88.3 |
| | 55 | 9.16 | 1.01 | 90.0 |

inhibitor molecule and mild steel [55–58]. The apparent activation energy ($E_a$) is calculated by Arrhenius equation [59],

$$\log(\mathrm{CR}) = \frac{-E_a}{2.303RT} + \log(A),\qquad(3.4)$$

where $E_a$ is the apparent activation energy and $A$ is the Arrhenius pre-exponential factor.

The $E_a$ values were obtained from the slope of Arrhenius plot as shown in figure 9 and the results are shown in table 9. It is obvious that $E_a$ values for the Q235 mild steel dissolution in 0.5 M H$_2$SO$_4$ solution

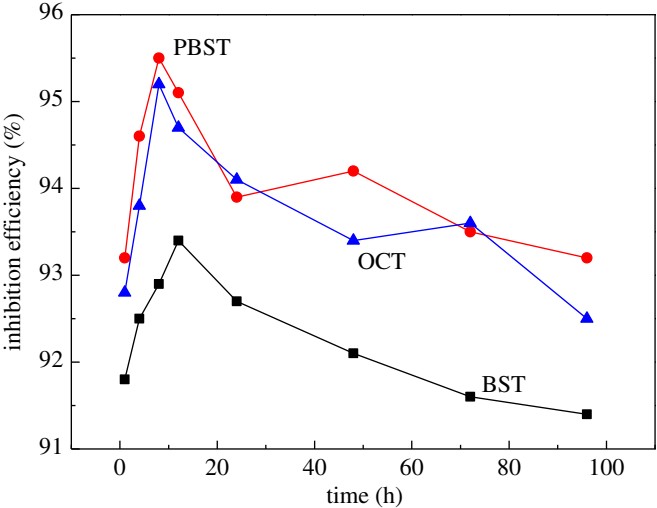

**Figure 10.** Effect of immersion time on the inhibition efficiency for 300 μM BST, PBST and OCT inhibitors onto Q235 mild steel in 0.5 M H$_2$SO$_4$ at 298 K.

**Table 9.** The apparent activation energy of mild steel corroded in 0.5 M H$_2$SO$_4$ without and with addition of 200 μM BST, PBST and OCT inhibitors.

| inhibitor | $E_a$ (KJ mol$^{-1}$) |
|---|---|
| blank | 11.1 |
| BST | 9.45 |
| PBST | 5.25 |
| OCT | 7.76 |

containing 200 μM inhibitors were smaller than that in the uninhibited. It was previously reported that the adsorption of organic inhibitor molecules follows chemisorptions mechanism when $E_a$ value was unchanged or lower compared to the blank [37,60], which was explained by some of the energy being consumed in the chemical reaction. This finding furthermore supports the conclusion that was inferred from the Langmuir isotherm that the adsorption behaviour of the synthesized compounds on mild steel surface conforms to chemisorptions.

## 3.8. Effect of immersion time

The impact of immersion time on the inhibition efficiency for 300 μM BST, PBST and OCT inhibitors onto Q235 mild steel in 0.5 M H$_2$SO$_4$ solution at 298 K is shown in figure 10. It can be seen that during the initial 8 h, the inhibition efficiency increased with immersion time for PBST and OCT inhibitor, whereas 12 h for BST inhibitor, which may be correlated to the film growth and rearrangement of the BST, PBST and OCT inhibitor molecules on Q235 surface. After that, the inhibition efficiency decreased with prolonging immersion time, which may be linked to desorption or dissolution of adsorbed inhibitor molecules [61]. It is noticeable that during the whole testing immersion time, the inhibition efficiency of PBST and OCT inhibitors is higher than that of BST inhibitor. Moreover, the inhibition efficiencies of all these inhibitors were still over 90% after 96 h immersion time, suggesting that these synthesized inhibitors were all long-term effective inhibitors for Q235 mild steel in 0.5 M H$_2$SO$_4$ solution.

## 3.9. Quantum chemical calculations

To explore the correlation between the inhibition behaviour and molecular structures, the quantum chemical calculation were performed, and the optimized geometry structures and the frontier

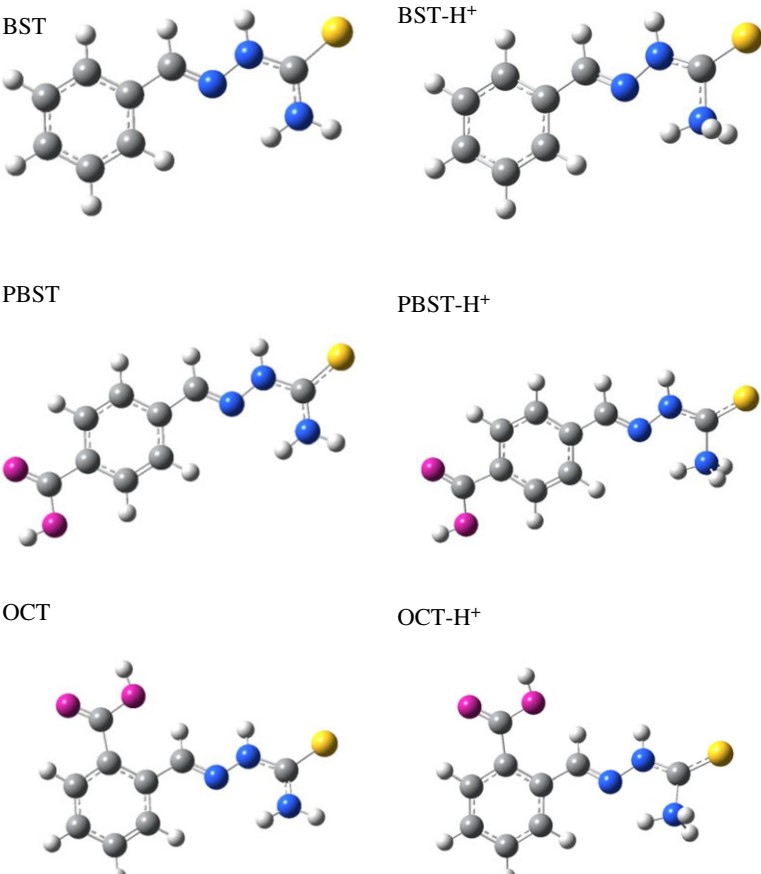

**Figure 11.** Optimized structure of BST, PBST and OCT and the protonated forms.

molecule orbital density distributions of these inhibitors as well as their protonated forms are presented in figures 11 and 12. The useful parameters, such as $E_{HOMO}$, $E_{LUMO}$, energy gap $\Delta E$ and dipole moment were determined and applied to explore the correlation between the inhibitor molecular structure and mild steel. According to the frontier molecular orbital theory, $E_{HOMO}$ is related to the ability of a molecule to donate electrons to appropriate electron acceptors, thus, a molecule exhibits stronger tendency to donate electrons to the steel vacancy d-orbital in the present study when the calculated $E_{HOMO}$ value is higher. Whereas, $E_{LUMO}$ corresponds to the electron accepting ability of the molecule, and a molecule has higher capability of accepting electrons when the calculated $E_{LUMO}$ value is lower [27,62,63]. Furthermore, the energy gap $\Delta E$ is an important parameter to evaluate the inhibitive effect of the inhibitor molecules. It was previously inferred [64,65] that an inhibitor possesses higher inhibition efficiency when its $\Delta E$ value is smaller, because lower energy is needed to remove an electron from the last occupied orbital.

According to Lukovits theorem [61], the value of ionization potential ($I$) and the electron affinity ($A$) can be derived from $E_{LUMO}$ and $E_{LUMO}$ by the following equations:

$$I = -E_{HOMO} \tag{3.5}$$

and

$$A = -E_{LUMO}. \tag{3.6}$$

Additionally, the absolute electronegativity ($\chi$), the global hardness ($\rho$) and softness ($\sigma$) of the inhibitor molecule are defined as follows:

$$\chi = \frac{I - A}{2}, \tag{3.7}$$

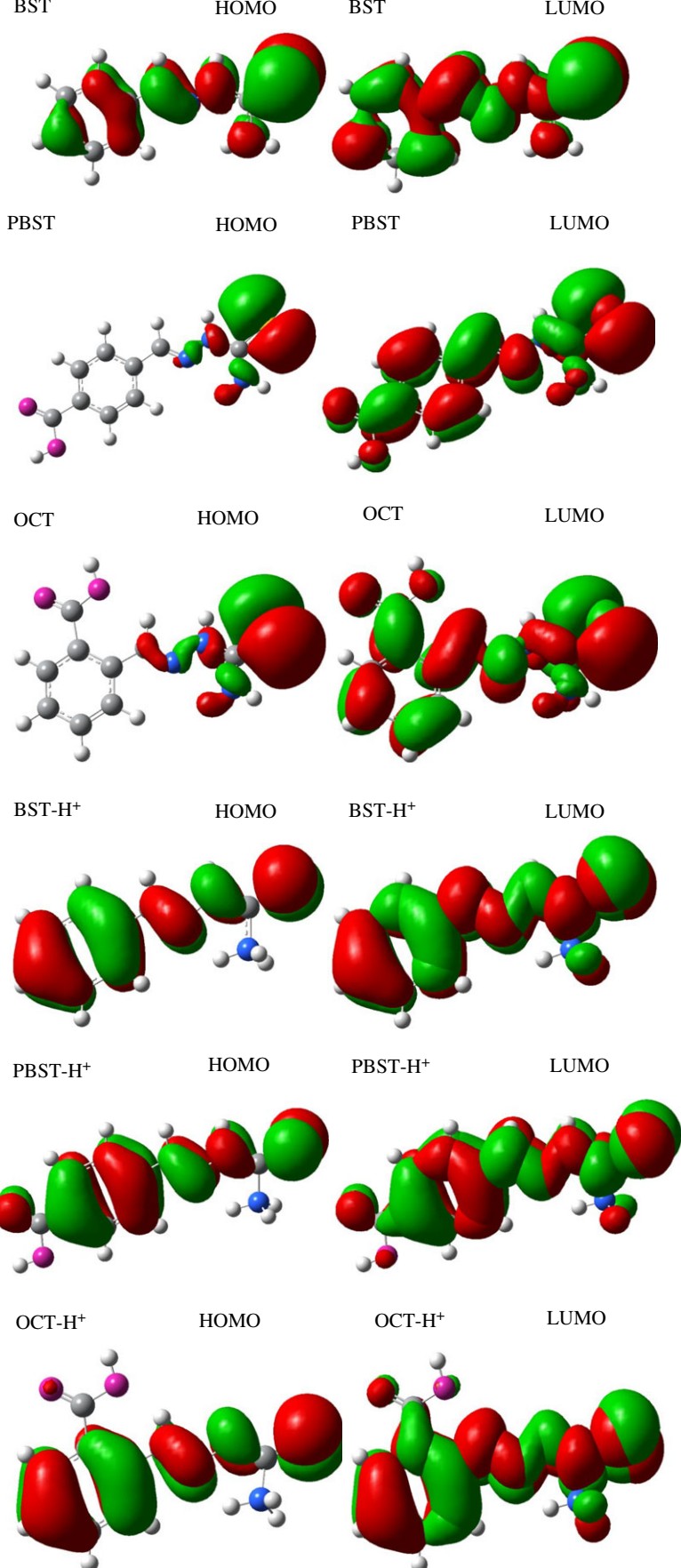

**Figure 12.** Frontier molecule orbital density distributions of these inhibitors and the protonated forms.

**Table 10.** Theoretical parameters of BST, PBST and OCT inhibitor and the protonated forms.

| parameters | BST | PBST | OCT | BST-H$^+$ | PBST-H$^+$ | OCT-H$^+$ |
|---|---|---|---|---|---|---|
| $E_{HOMO}$ (eV) | −5.9772 | −6.1944 | −6.0986 | −10.1295 | −10.3085 | −10.3322 |
| $E_{LUMO}$ (eV) | −2.1490 | −2.7433 | −2.5512 | −6.2983 | −6.5176 | −6.4123 |
| $\Delta E$ (eV) | 3.8282 | 3.4511 | 3.5474 | 3.8312 | 3.7909 | 3.9199 |
| $\mu$ (D) | 5.4560 | 3.1922 | 4.3649 | 4.3037 | 10.8639 | 9.4524 |
| $I$ (eV) | 5.9772 | 6.1944 | 6.0986 | 10.1295 | 10.3085 | 10.3322 |
| $A$ (eV) | 2.1490 | 2.7433 | 2.5512 | 6.2983 | 6.5176 | 6.4123 |
| $\chi$ (eV) | 4.0631 | 4.4689 | 4.3249 | 8.2139 | 8.4131 | 8.3722 |
| $\rho$ (eV) | 1.9141 | 1.7256 | 1.7737 | 1.9156 | 1.8954 | 1.9599 |
| $\sigma$ [(eV)$^{-1}$] | 0.5224 | 0.5795 | 0.5638 | 0.5220 | 0.5276 | 0.5102 |

$$\rho = \frac{I-A}{2} \tag{3.8}$$

and

$$\sigma = \frac{1}{\rho}. \tag{3.9}$$

All of these calculated quantum chemical parameters are summarized in table 10. It is apparent that the neutral form of PBST inhibitor as well as its protonated form PBST-H$^+$ has a minimum value of $E_{LUMO}$ and lowest value of $\Delta E$ which is well in agreement with its highest inhibition efficiency. Moreover, PBST inhibitor shows the lowest value of dipole moment ($\mu$), which will favour accumulation of the inhibitor [66]. However, there is still a controversy about the correlation between the dipole moment and inhibition efficiency that many researchers suggested that the inhibition efficiency increased with the increase of dipole moment [67,68], while others stated that inhibitor molecule with a lower value of dipole moment revealed higher inhibition efficiency [66]. Generally, the value of electronegativity $\chi$ represents the chemical potential and a higher value indicates better inhibition performance. In addition, an inhibitor always shows a higher inhibition efficiency when the value of global hardness is smaller according to the hard-soft acid base (HSAB) principle [69]. Inspection of table 10 also demonstrates that PBST has the highest electronegativity and lowest global hardness, resulting in the maximum inhibition efficiency compared to the other two inhibitors, which is consistent with the result of weight loss and electrochemical measurements.

# 4. Conclusion

Gravimetric measurements, polarization curves, electrochemical impedance spectroscopy and scanning electron microscopy (SEM) were used to study the inhibition behaviour of three new benzaldehyde thiosemicarbazone derivatives for mild steel in 0.5 M $H_2SO_4$ solution. Results revealed that all these compounds are good inhibitors for Q235 steel in 0.5 M $H_2SO_4$ solution and PBST inhibitor showed the maximum inhibition efficiency of 96.6% at 300 μM. The inhibition efficiency increases with increasing inhibitors concentration and temperature. The results of polarization curves indicated that these three compounds behaved as mixed type and PBST and OCT predominantly anodic. The adsorption of these inhibitors on Q235 steel surface was according to Langmuir adsorption isotherm. The results of theoretical calculation and SEM studies were found to be in good agreement with that of weight loss and electrochemical measurements.

Data accessibility. The datasets generated in this work are available from the corresponding author on reasonable request. Our data are available from the Dryad Digital Repository: https://doi.org/10.5061/dryad.7cn76k3 [70].
Authors' contributions. H.H.Z. performed the measurements. Y.C. wrote the manuscript. Z.Z. supervised and interpreted the research. C.K.Q. helped us revise the manuscript and performed the theoretical calculations. All authors discussed the results and commented on the manuscript. All authors have read and approved the manuscript before submission.
Competing interests. The authors declare no competing interests.

Funding. The authors wish to acknowledge the financial support of the National Natural Science Foundation of China (Project 21403194, 51771173), the Natural Science Foundation of Shandong Province (ZR2019QEM003, ZR2014BQ027, ZR2016BP11), the Project of Focus on Research and Development Plan in Shandong Province (2019GSF111048), the Major Project of Binzhou University (2017ZDL02) and Scientific Research Fund of Binzhou University (BZXYLG1904, BZXYFB20140805, BZXYL1403).

Acknowledgements. The authors would like to thank Zhongnian Yang, Yuanwei Liu and Junying Yin (Binzhou University) for their kind help.

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
