## [Reviewer comments · Royal Society Open Science]

Review History

RSOS-190192.R0 (Original submission)

Review form: Reviewer 1

Is the manuscript scientifically sound in its present form?

Yes

Are the interpretations and conclusions justified by the results?

Yes

Is the language acceptable?

Yes

Is it clear how to access all supporting data?

Not Applicable

Do you have any ethical concerns with this paper?

No

Have you any concerns about statistical analyses in this paper?

No

Recommendation?

Accept with minor revision (please list in comments)

Comments to the Author(s)

In this paper, the inhibition behavior of three new benzaldehyde thiosemicarbazone derivatives on mild steel in 0.5 M H₂SO₄ solution was investigated using gravimetric measurements, potentiodynamic polarization, electrochemical impedance spectroscopy (EIS), scanning electron microscopy and theoretical calculations. The inhibitor exhibited good inhibition efficiency (96.6%). The work is interesting and I feel that it could be published after making the following corrections/ suggestions.

1. "Acid corrosion" should be added as one of the keyword.
2. In abstract, the sentence "The analysis of polarization curves indicated that the three benzaldehyde thiosemicarbazone derivatives acted were of mixed type and PBST and OCT predominantly anodic." should be rewritten.
3. Throughout the text, "molecule structure" should be replaced by "molecular structure".
4. Page 5, an " η " is missing from equation (5).

Review form: Reviewer 2

Is the manuscript scientifically sound in its present form?

Yes

Are the interpretations and conclusions justified by the results?

Yes

Is the language acceptable?

No

Do you have any ethical concerns with this paper?

No

Recommendation?

Major revision is needed (please make suggestions in comments)

Comments to the Author(s)

Title: Inhibition behavior of mild steel by three new benzaldehyde thiosemicarbazone derivatives in 0.5 M H₂SO₄: Experimental and computational study

Journal: Royal Society Open Science

Manuscript number Review: RSOS-190192

Reviewer:

- 1- The English of the manuscript is adequate for its publication but there are some corrections to be made.
- 2- The introduction is poorly structured.
- 3- The solubility of inhibitor in the solvent must be explained. The effect of the solvent on the inhibition needs more discussion.
- 4- In the gravimetric method why did you choose 8 hours as immersion time.

- 5- It seems that the recommended scan rate for obtaining Tafel slopes is lower than 1 mV s⁻¹ (~0.2 V s⁻¹). Why was the reason to use 1 mV s⁻¹?
 - 6- The figures are not clear?
 - 7- The E_{corr} unit must be corrected
 - 6- The OCP vs time should be shown and discussed.
 - 7- The authors used curve fitting or Tafel extrapolation method in obtaining the values of corrosion currents??
 - 8- Thank you to give a detailed discussion on the Tafel slopes (ba and bc).
 - 9- Nyquist plots should show some define frequencies.
 - 10-
 - 11- A representative of example simulation of Nyquist and Bode diagrams with suggested models without and with of inhibitor should be given. It would be necessary to give, in the Table 5, the scattering (error bars) of all fitted parameters to evaluate the accuracy of the used models.
 - 12- The author must justify the choice of use the CPE.
 - 13- In part electrochemical impedance spectroscopy measurements: the authors should give and interpret in detail the variation of n value in the absence and presence of inhibitory molecule.
 - 14- You must add the error for each parameter in the impedance tables.
 - 15- The sentence "The negative value of indicates that the $\Delta 0$ adsorption of benzaldehyde thiosemicarbazone derivatives onto Q235 mild steel in 0.5 M H₂SO₄ solution is spontaneous (due the negative value!!)." is wrong. Please delete this. This parameter is a thermodynamic representation of the equilibrium constant, that is all.
 - 16- Why you did not calculate the other activation parameters?
 - 17- You must add all the equations concerning the theoretical part.
 - 18- The paper includes a rather verbose discussion of DFT calculated molecular electronic properties using HOMO-LUMO type cliché inferences that have been used countless times in the literature. Such an approach is not acceptable in the year 2018, because the DFT methodology to explicitly model the adsorption of molecules on materials surfaces is well established and mature for almost two decades (e.g., see Hammer & Norskov, *Advances in Catalysis* 45 (2000) 71-129). It has been recently clearly demonstrated that these HOMO-LUMO parameters are not very useful, because for a large data set of corrosion inhibitors the purported correlations between inhibitor's frontier molecular orbital parameters and inhibitor efficiency disappears, see: *J. Mater. Chem. A* 2 (2014), 16660-16668, *Green Chem.* 16 (2014), 3349-3357. Please refer to *Corros. Sci.* 85 (2014) 109-114, stating: "In terms of predictive power, such an approach has at best limited value, and is potentially simply misleading..."
 - 19- It is well known that the benzaldehyde thiosemicarbazone derivatives inhibitors could be protonated in acid solution, so the quantum chemical calculation should be considered the protonated inhibitors.
 - 20- Dynamic molecular simulation needs to be added?
- Based on my observations, I would recommend this manuscript for publication in "Royal Society Open Science" but after the previous major corrections.

Decision letter (RSOS-190192.R0)

01-Jul-2019

Dear Dr Chen:

Title: Inhibition behavior of mild steel by three new benzaldehyde thiosemicarbazone derivatives in 0.5 M H₂SO₄: Experimental and computational study
Manuscript ID: RSOS-190192

Thank you for submitting the above manuscript to Royal Society Open Science. On behalf of the Editors and the Royal Society of Chemistry, I am pleased to inform you that your manuscript will be accepted for publication in Royal Society Open Science subject to minor revision in accordance with the referee suggestions. Please find the reviewers' comments at the end of this email.

The reviewers and handling editors have recommended publication, but also suggest some minor revisions to your manuscript. Therefore, I invite you to respond to the comments and revise your manuscript. I apologise that this has taken longer than usual.

Please also include the following statements alongside the other end statements. As we cannot publish your manuscript without these end statements included, if you feel that a given heading is not relevant to your paper, please nevertheless include the heading and explicitly state that it is not relevant to your work. We have included a screenshot example of the end statements for reference.

- Acknowledgements

- Funding statement

Please include a funding section after your main text which lists the source of funding for each author.

Because the schedule for publication is very tight, it is a condition of publication that you submit the revised version of your manuscript before 10-Jul-2019. Please note that the revision deadline will expire at 00.00am on this date. If you do not think you will be able to meet this date please let me know immediately.

- 1) A text file of the manuscript (tex, txt, rtf, docx or doc), references, tables (including captions) and figure captions. Do not upload a PDF as your "Main Document".
- 2) A separate electronic file of each figure (EPS or print-quality PDF preferred (either format should be produced directly from original creation package), or original software format)
- 3) Included a 100 word media summary of your paper when requested at submission. Please ensure you have entered correct contact details (email, institution and telephone) in your user account
- 4) Included the raw data to support the claims made in your paper. You can either include your data as electronic supplementary material or upload to a repository and include the relevant doi within your manuscript

5) All supplementary materials accompanying an accepted article will be treated as in their final form. Note that the Royal Society will neither edit nor typeset supplementary material and it will be hosted as provided. Please ensure that the supplementary material includes the paper details where possible (authors, article title, journal name).

Best wishes,
Dr Laura Smith
Publishing Editor, Journals

On behalf of the Subject Editor Professor Anthony Stace and the Associate Editor Professor Tobias Hertel.

RSC Associate Editor:
Comments to the Author:
(There are no comments.)

RSC Subject Editor:
Comments to the Author:
(There are no comments.)

Reviewer comments to Author:
Reviewer: 1

Comments to the Author(s)

In this paper, the inhibition behavior of three new benzaldehyde thiosemicarbazone derivatives on mild steel in 0.5 M H₂SO₄ solution was investigated using gravimetric measurements, potentiodynamic polarization, electrochemical impedance spectroscopy (EIS), scanning electron microscopy and theoretical calculations. The inhibitor exhibited good inhibition efficiency (96.6%). The work is interesting and I feel that it could be published after making the following corrections/ suggestions.

1. "Acid corrosion" should be added as one of the keyword.

2. In abstract, the sentence “The analysis of polarization curves indicated that the three benzaldehyde thiosemicarbazone derivatives acted were of mixed type and PBST and OCT predominantly anodic.” should be rewritten.
3. Throughout the text, “molecule structure” should be replaced by “molecular structure”.
4. Page 5, an “ η ” is missing from equation (5).

Reviewer: 2

Comments to the Author(s)

Title: Inhibition behavior of mild steel by three new benzaldehyde thiosemicarbazone derivatives in 0.5 M H₂SO₄: Experimental and computational study

Journal: Royal Society Open Science

Manuscript number Review: RSOS-190192

Reviewer:

- 1- The English of the manuscript is adequate for its publication but there are some corrections to be made.
- 2- The introduction is poorly structured.
- 3- The solubility of inhibitor in the solvent must be explained. The effect of the solvent on the inhibition needs more discussion.
- 4- In the gravimetric method why did you choose 8 hours as immersion time.
- 5- It seems that the recommended scan rate for obtaining Tafel slopes is lower than 1 mV s⁻¹ (~0.2 V s⁻¹). Why was the reason to use 1 mV s⁻¹?
- 6- The figures are not clear?
- 7- The E_{corr} unit must be corrected
- 6- The OCP vs time should be shown and discussed.
- 7- The authors used curve fitting or Tafel extrapolation method in obtaining the values of corrosion currents??
- 8- Thank you to give a detailed discussion on the Tafel slopes (ba and bc).
- 9- Nyquist plots should show some define frequencies.
- 10-
- 11- A representative of example simulation of Nyquist and Bode diagrams with suggested models without and with of inhibitor should be given. It would be necessary to give, in the Table 5, the scattering (error bars) of all fitted parameters to evaluate the accuracy of the used models.
- 12- The author must justify the choice of use the CPE.
- 13- In part electrochemical impedance spectroscopy measurements: the authors should give and interpret in detail the variation of n value in the absence and presence of inhibitory molecule.
- 14- You must add the error for each parameter in the impedance tables.
- 15- The sentence “The negative value of indicates that the $\Delta 0$ adsorption of benzaldehyde thiosemicarbazone derivatives onto Q235 mild steel in 0.5 M H₂SO₄ solution is spontaneous (due the negative value!!).” is wrong. Please delete this. This parameter is a thermodynamic representation of the equilibrium constant, that is all.
- 16- Why you did not calculate the other activation parameters?
- 17- You must add all the equations concerning the theoretical part.
- 18- The paper includes a rather verbose discussion of DFT calculated molecular electronic properties using HOMO-LUMO type cliché inferences that have been used countless times in the literature. Such an approach is not acceptable in the year 2018, because the DFT methodology to explicitly model the adsorption of molecules on materials surfaces is well established and mature for almost two decades (e.g., see Hammer & Norskov, *Advances in Catalysis* 45 (2000) 71-129). It has been recently clearly demonstrated that these HOMO-LUMO parameters are not very useful, because for a large data set of corrosion inhibitors the purported correlations between inhibitor's frontier molecular orbital parameters and inhibitor efficiency disappears, see: *J. Mater. Chem. A* 2

(2014), 16660-16668, Green Chem. 16 (2014), 3349-3357. Please refer to Corros. Sci. 85 (2014) 109-114, stating: "In terms of predictive power, such an approach has at best limited value, and is potentially simply misleading..."

19- It is well known that the benzaldehyde thiosemicarbazone derivatives inhibitors could be protonated in acid solution, so the quantum chemical calculation should be considered the protonated inhibitors.

20- Dynamic molecular simulation needs to be added?

Based on my observations, I would recommend this manuscript for publication in "Royal Society Open Science" but after the previous major corrections.

Author's Response to Decision Letter for (RSOS-190192.R0)

See Appendix A.

Decision letter (RSOS-190192.R1)

22-Jul-2019

Dear Dr Chen:

Title: Inhibition behavior of mild steel by three new benzaldehyde thiosemicarbazone derivatives in 0.5 M H₂SO₄: Experimental and computational study

Manuscript ID: RSOS-190192.R1

It is a pleasure to accept your manuscript in its current form for publication in Royal Society Open Science. The chemistry content of Royal Society Open Science is published in collaboration with the Royal Society of Chemistry.

On behalf of the Subject Editor Professor Anthony Stace and the Associate Editor Professor Tobias Hertel.

RSC Associate Editor
Comments to the Author:
(There are no comments.)

Reviewer(s)' Comments to Author: